# Therapeutical Options in ROS1—Rearranged Advanced Non Small Cell Lung Cancer

**DOI:** 10.3390/ijms241411495

**Published:** 2023-07-15

**Authors:** Brigida Stanzione, Alessandro Del Conte, Elisa Bertoli, Elisa De Carlo, Alberto Revelant, Michele Spina, Alessandra Bearz

**Affiliations:** 1Department of Medical Oncology, Centro di Riferimento Oncologico di Aviano (CRO), IRCCS, 33081 Aviano, Italy; brigida.stanzione@cro.it (B.S.); alessandro.delconte@cro.it (A.D.C.); elisa.bertoli@cro.it (E.B.); elisa.decarlo@cro.it (E.D.C.); mspina@cro.it (M.S.); 2Department of Medicine (DAME), University of Udine, 33100 Udine, Italy; 3Department of Radiotherapy, Centro di Riferimento Oncologico di Aviano (CRO), IRCCS, 33081 Aviano, Italy; alberto.revelant@cro.it

**Keywords:** ROS1, non small cell lung cancer, target therapy

## Abstract

ROS proto-oncogene 1 (ROS1) rearrangements occur in 0.9–2.6% of patients with non small cell lung cancer (NSCLC), conferring sensitivity to treatment with specific tyrosine-kinase inhibitors (TKI). Crizotinib, a first-generation TKI, was the first target-therapy approved for the first-line treatment of ROS1-positive NSCLC. Recently, entrectinib, a multitarget inhibitor with an anti-ROS1 activity 40 times more potent than crizotinib and better activity on the central nervous system (CNS), received approval for treatment-naive patients. After a median time-to-progression of 5.5–20 months, resistance mechanisms can occur, leading to tumor progression. Therefore, newer generation TKI with greater potency and brain penetration have been developed and are currently under investigation. This review summarizes the current knowledge on clinicopathological characteristics of ROS1-positive NSCLC and its therapeutic options.

## 1. Introduction

Lung cancer is considered the first cause of death from cancer worldwide [1]. Non-small cell lung cancer (NSCLC), which accounts for around 84% of all lung cancer cases and is frequently diagnosed at an advanced stage, has a poor prognosis and low survival rates. NSCLC is not a single entity but rather represents a variety of distinct illnesses, based on its molecular characteristics and, in some circumstances, on the expression of particular targetable oncogenic drivers [2]. ROS1 is a tyrosine kinase receptor belonging to the insulin receptor family. Its gene is on chromosome 6. It is made up of an extracellular N-terminal domain, a single trans-membrane domain, and an intracellular C-terminal region that contains the kinase domain [3,4]. Wild-type ROS1′s role is not completely known, but it seems to be involved in differentiation of epithelial tissues during embryogenesis. Chromosomal rearrangements involving ROS1 were first identified in glioblastoma, but they have been identified also in cholangiocarcinoma, gastric cancer, ovarian cancer, soft-tissue sarcomas, breast cancer, lung cancer and many other tumor types [5]. Several gene fusion partners have been discovered. The most frequent fusion gene is CD74–ROS1, representing 44% of cases, followed by EZRs–ROS1, SDC4–ROS1 and SLC34A2. The upregulation of the SHP-2 phosphatase, MAPK/ERK pathway, PI3K/AKT/mTOR pathway, and JAK/STAT pathway, which control cellular survival, growth, and proliferation, is a characteristic of all fusion genes. These genes are oncogenic and have constitutive, ligand-independent catalytic activity [6]. The prognostic role of these different fusion genes has been explored in several studies but clear conclusions cannot be outlined. Interestingly, oncogenic co-mutations can be found in 36% of ROS1 positive NSCLC, in particular EGFR or KRAS mutations, MET amplification or ALK translocation [7]. ROS1 rearrangements are present in 0.9–2.6% of NSCLCs. ROS1-positive NSCLCs share some clinicopathological characteristics with ALK-positive NSCLCs, although they are different entities. In fact, ROS1 rearrangements are more frequently found in adenocarcinomas (with a predominance of solid, papillary, acinar, cribriform, and mucinous histology patterns), women, young patients and light- or never-smokers. Furthermore, ROS1-positive tumors are usually diagnosed in an advanced stage (III-IV), with brain metastases and lymph-node involvement; they are also associated with major thromboembolic risk, included trombotic microangiopathy or disseminated intravascular coagulation (DIC) [8,9,10,11,12,13,14]. Central nervous system (CNS) involvement is frequent not only at diagnosis (36% of patients), but also as the first-site of progression (47% of patients) [11]. All metastatic lung cancers should be tested for the presence of ROS1 rearrangements, regardless of clinical characteristics. In particular, ROS1 should be sought not only in adenocarcinomas, but also in tumors with mixed histology or squamous-cell carcinomas, because an adenocarcinoma component cannot be excluded, especially in never-smokers or younger (<50 years) patients [15]. The techniques that can be used to detect the presence of ROS1 are immunohistochemistry (IHC), fluorescence in situ hybridation (FISH), reverse-transcriptase-polymerase-chain-reaction (RT-PCR) and next generation sequencing (NGS). These techniques are not always available in all cancer treatment centers, but the use of them is essential for a correct diagnosis and an optimal setting of the therapeutic path. IHC is usually used as a screening technique [16,17]. It has lower costs than other techniques, and a high sensitivity, but a variable specificity, ranging from 70% to 90%. Its limits are the lack of globally accepted scores, the difficulty in interpretation of results and the need to use other techniques to confirm results, in the case of positivity or questionable results. In particular, a weak ROS1 expression can be found in hyperplastic type II pneumocytes, alveolar macrophages and osteoclasts, making the evaluation of the outcome more difficult. IHC false-positives are more common in EGFR-mutated lepidic or acinar adenocarcinomas. Actually, there are three commercially available anti-ROS1 antibodies. When a positive IHC is found in contrast with a negative FISH, due to the presence of other oncogenic driver mutations, the use of a further technique is necessary, usually NGS. The gold standard to identify ROS1 rearrangements is represented by FISH, thanks to its high specificity and sensitivity [18]. FISH testing can be performed either on histological sections or cytological specimens; it requires low input of material and it is possible to have results in a short time. When more than 15% of the cells display separation of the 3’ and 5’ probes or a distinct 3’ signal (centromeric), the sample is deemed positive. Its limits are the need of an experienced pathologist due to the difficulty level to interpret the results, impossibility to know ROS1 fusion partners, and need of sufficient amount of tumor cells (more than 50) [19]. RT-PCR, despite its high sensitivity and specificity, is rarely used for its high costs and technical difficulties. In particular, this technique requires different steps, such as RNA extraction, complementary DNA synthesis, quantitative PCR and analysis, with a high risk of variations [20,21,22]. NGS allows to test simultaneously many predictive biomarkers, with high sensitivity and specificity [23,24]. It is possible to identify several ROS1 fusion partners as well as other oncogenic molecular targets, by using tumor DNA or RNA. NGS can be performed on tumor tissue or plasma. The cons of NGS are long lead times and high costs, limiting its use in routine clinical practice, especially in small hospital centers. ROS1 turned out like a driver for targeted therapies, changing the natural history of this disease. Different tyrosine kinase inhibitors are now available in ROS1-rearranged NSCLC, and several clinical trials are ongoing in this setting [25,26]. In Figure 1, the timeline of main discoveries in ROS1-rearranged NSCLC is reported, from the first identification of ROS1-rearrangements to the most recently introduced drugs.

## 2. Crizotinib

Several retrospective and prospective trials demonstrated the efficacy and safety of crizotinib, a first-generation multitarget oral tyrosine kinase inhibitor, in ROS1-positive NSCLC (Table 1). A phase I trial, PROFILE 1001, enrolled 53 patients to receive crizotinib at the dose of 250 mg twice a day. In this trial, most patients had previously received one or two lines of therapy (42% and 44% of cases, respectively) and had been predominantly exposed to a platinum-based chemotherapy (80%) [27]. The updated analysis, after a median follow-up period of 62.6 months, showed an overall response rate (ORR) of 72% (95% CI 58–83%), a median progression-free survival (mPFS) of 19.3 months (95% CI 15.2–39.1 months), and a median overall survival (mOS) of 51.4 months (95% CI 29.3 months—NR) [28]. Crizotinib is five times more effective at blocking ROS1 than ALK, according to the ROS1-specific structure, which explains why ROS1-positive NSCLC experienced greater responses than ALK-positive ones. With 94% of adverse events being categorized as minor (grade 1 or 2), the safety profile was satisfactory. The most frequent side effects associated with treatment included peripheral edema (47%), diarrhea (45%), nausea (51%), and vision impairment (87%). In the EUCROSS study, a multicentric phase II single-arm prospective trial evaluating the role of crizotinib in ROS1-positive NSCLC, ORR was 70% (95% CI 51–85%), while mPFS was 20 months (95% CI 8 months-NR), confirming not only PROFILE 1001 data, but also results derived from an East Asian prospective phase II trial [29]. Noteworthy is METROS study, designed with the aim to evaluate the activity and safety of crizotinib in pretreated patients with NSCLC and ROS1 rearrangements or MET amplifications or mutations [30]. The ORR and mPFS for this trial were 65% (95% CI 44–82%) and 23 months (95% CI 15–30 months), respectively. A recent update from the Acsè Program, a basket trial on crizotinib for patients with solid tumors who have alterations of MET, ALK, or ROS-1, showed an ORR of 47.2% with a best response rate reaching 69.4% and median PFS and OS, respectively, at 5.5 months (95% CI 4.2–9.1 months) and 17.2 months (95% CI 6.8–32.8 months) [31]. These data are less favorable than results derived from the PROFILE 1001 trial, probably because patients were more heavily pre-treated and had a worse Eastern Cooperative Oncology Group performance score (PS). Most studies showed a worst prognosis for patients with brain metastases, with a shorter mPFS, probably due to limited intracranial activity for crizotinib. There are no trials comparing crizotinib and chemotherapy in patients with advanced NSCLC and ROS1 rearrangements, although some evidence suggests a particular activity of pemetrexed-based chemotherapy [32]. The efficacy demonstrated by crizotinib and its acceptable toxicity profile is responsible for its approval in the first-line treatment of NSCLC harboring ROS1 rearrangement.

As for other oncogenic-driven NSCLCs, also ROS1-positive NSCLC can present de-novo or acquired resistance to crizotinib [33]. Primary resistance to crizotinib can be due to the development of KRAS mutations, a limited CNS penetration, or BIM deletion polymorphisms [34,35]. On the other hand, mechanisms of acquired resistance can be ROS1-dependent or ROS1–independent. In the first group, several ROS1 mutations have been described, leading to steric interference with drug binding, preventing its access to the enzyme active site or increasing kinase activity [36]. G2032R occurs in 41% of cases and is the most common mutation; D2033N, described in 6% of cases, confers in vitro and in vivo sensitivity to cabozantinib; S1986F or S1986Y, described in 6% of cases, showed sensitivity to lorlatinib and resistance to crizotinib and ceritinib; L2026M confers sensitivity to lorlatinib, repotrectinib and foretinib [36,37,38,39,40,41,42,43]. Among ROS1-independent mechanisms of resistance, in vivo and in vitro studies have reported acquisition of an activating KIT mutation, the activation of an epithelial-to-mesenchymal transition (EMT), a switch in the control of growth and survival from ROS1 to EGFR, and transformation into small cell lung cancer (SCLC) or KRAS amplifications [44,45,46,47]. The phenotype change in SCLC seems to be associated with retinoblastoma-1 (RB-1) and TP-53 genes inactivation and loss of ROS1-fusion expression and is associated with less sensitivity to platinum-etoposide chemotherapy [25]. Currently, new-generation TKIs have been developed to overcome the main resistance mechanisms.

## 3. Entrectinib

Entrectinib is a multitarget inhibitor, able to bind ROS1, ALK, and pan-tropomyosin-receptor kinase (pan-TRK). Entrectinib can pass the blood-brain barrier and seems to be 40 times more potent than crizotinib [48,49]. It is administered at the dose of 600 mg once a day. Four clinical studies (STARTRK-2, ALKA-372–001, STARTRK-1 and STARTRK-NG) demonstrated the efficacy and safety of entrectinib in patients with ROS1 rearrangements [50,51,52,53]. Table 2 reports the main results of the described studies. Drilon et al. first reported the results of the phase I/II studies ALKA-372-001 and STARTRK-1, which were subsequently supplemented by the phase 2 study STARTRK-2 data [51]. An updated analysis evaluated 161 patients; most of the patients had an ECOG performance status of 0–1 (90.1%), were non-smokers (62.7%), and pretreated (62.8%); 34.8% had brain metastases [52]. After a median follow-up of 15.8 months, the ORR was 67.1% (95% CI 59.3–74.3%), the median time to response (mTTR) was 0.95 months (95% CI 0.7–26.6 months). Median time to progression (mTTP) was 2.8 months (95% CI 0.4–21.1 months); mPFS was 15.7 months (95% CI 11.0–21.1 months), while mOS was not reached. To evaluate the efficacy of entrectinib on brain metastases, an analysis of data from patients with brain metastases was performed. In this subgroup, the ORR was 62.5%, the intracranial ORR 52.2% (95% CI, 37.0–67.1%), while the median intracranial PFS was 8.3 months (95% CI, 6.4–15.7 months). The majority of treatment-related adverse events were grade 1–2; the most frequent are represented by dysgeusia (42.9%), malaise (34.3%), and constipation (31.4%). Other notable adverse events were weight gain (8.1%), ALT elevation (3.3%), and diarrhea (2.9%). In the updated analysis, only seven patients (3.3%) reported grade 4 adverse events. The phase I STARTRK-NG study, exploring the role of entrectinib in young adults, reported similar results [53]. Preclinical data show that entrectinib has no activity when ROS1 G2032R and L2026M mutations occur, endorsing its role in first-line treatment and confirming its failure in the case of acquired crizotinib resistance [54].

## 4. Entrectinib vs. Crizotinib

Crizotinib and entrectinib are being compared head-to-head in patients with ROS1-positive non small cell lung cancer, including those who have brain metastases, in a phase III randomized controlled trial (NCT04603807) that has been accepting patients since 2021 [55]. The comparative efficacy and safety of the two medications in a clinical trial environment will be directly demonstrated by this investigation. Until the results of this study are available, the only feasible assessments can only be derived from indirect comparisons. In particular, Tremblay et al. published a paper in which they produced a clinical trial-to-clinical trial simulated treatment comparison (STC) using data from the regulatory approval-supporting trials for crizotinib and entrectinib to compare the efficacy of crizotinib and entrectinib in ROS1-positive non small cell lung cancer [56]. Obviously, this type of analysis is affected by numerous biases and high variability. Although the crizotinib and entrectinib study populations were similar in many trial inclusion/exclusion criteria, some crucial differences may impact the outcome of the indirect comparison. The population of the studies PROFILE 1001 and STARTRK-1, STARTRK-2 and ALKA-372-001 presented substantial differences, particularly with regard to sex, ECOG PS and smoking habit. In particular, PROFILE 1001 included patients with ECOG PS 0–1, while ECOG PS 2 patients were only included under special circumstances. On the contrary, patients in the entrectinib trials had an ECOG PS of 0–2, indicating that some of the participants may have had lower performance status. According to estimates, results are much poorer when ECOG PS is higher. In addition, PROFILE 1001 had a much longer follow-up time as well. Another drawback is that the research involved comparatively few patients, and results were measured differently. Furthermore, data on the presence of CNS metastases were not collected at baseline in the PROFILE 1001 study, in contrast to the entrectinib registrational study. With the above limits, Tremblay et al. found that crizotinib showed nonsignificant ORR advantages over entrectinib both before and after adjustment for key variables. When compared to entrectinib, crizotinib was linked with non-significant but prolonged mPFS and mDOR both before and after correction. The 12-month OS after adjustment showed a similar pattern, despite the fact that the uncorrected analysis indicated a non-significantly higher risk of mortality for crizotinib. Chu et al. published a similar clinical trial-to-clinical trial comparison that included entrectinib pivotal data from the ALKA-372-001, STARTRK-1, and STARTRK-2 studies in ROS1-positive NSCLC and matching-adjusted indirect comparison (MAIC) of crizotinib from PROFILE 1001 and individual patient data (IPD) for entrectinib [57]. This MAIC discovered that entrectinib may significantly outperform crizotinib in terms of ORR results but found no statistically significant difference between the two drugs’ PFS or OS outcomes. The differences between these indirect comparisons are the evaluation of the frequency of brain metastases, which is present only in the MAIC, and the type of data used; the MAIC used an earlier crizotinib and entrectinib data cut, whereas STC evaluated the updated OS from PROFILE 1001 and a larger sample size with at least 6 months’ follow-up from the integrated entrectinib analysis. While awaiting the data related to the phase III trial with the face-to-face comparison of the two drugs, in daily clinical practice, it could be hypothesized that crizotinib is preferred in the absence of brain metastases and entrectinib prescribed in the case of CNS localizations, exploiting its greater ability to penetrate through the blood-brain barrier. Table 3 summarizes outcomes of the main trials of crizotinib and entrectinib in ROS1-positive NSCLC.

## 5. Ceritinib

Ceritinib is a second-generation ALK/ROS1 TKI, able to overcome crizotinib resistance only when specific mutations occur, such as L2026M, M2001T, and G2101A [58]. In a phase II prospective trial, Ceritinib was administered at the dose of 750 mg/die to both crizotinib-naive and crizotinib-pretreated patients [59]. ORR was 62% (95% CI 45–77%), with a median PFS of 9.3 months (95% CI 0–22 months). In the subgroup of crizotinib-naïve patients, PFS was 19.3 months (95% CI 1–37 months), while in the subgroup of patients with brain metastases, the brain ORR was 63% (95% CI 31–86%). Grade 3–4 adverse events were reported in 37% of the population. Currently, ceritinib is not approved for the first-line treatment, but it could have a future role after progression to crizotinib.

## 6. Lorlatinib

Lorlatinib is a third-generation TKI that targets the ALK and ROS1 kinase domains and is reversible, highly selective, and potent. It can cross the blood-brain barrier thanks to a reduced efflux caused by P-glicoprotein-1 [39,60,61]. In preclinical studies, it has shown an in vitro activity against ROS1 L2026M, D2033 and S1986Y/F, while it does not work against the most common ROS1 mutations. In a phase I trial followed by a phase I/II trial, lorlatinib was administered at the dose of 100 mg once-a-day to 21 TKI treatment-naïve patients and 40 crizotinib-treated patients, diagnosed with a ROS1-positive NSCLC [62,63,64,65]. In the TKI naïve subgroup, ORR was 62% (95% CI, 38–82%), mPFS was 21 months (95% CI, 4.2–31.9 months), and brain ORR was 64% (95% CI 31–89%). The good activity against brain metastases is likely the reason why the median brain PFS was not reached. The intracerebral ORR, median PFS, and ORR in the subgroup that had previously received crizotinib were, respectively, 35% (95% CI 21–52%), 8.5 months (95% CI 4.7–15.2 months), and 50% (95% CI 29–71%). Hypercholesterolemia (65%), hypertriglyceridemia (42%), peripheral edema (39%), peripheral neuropathies, (26%), changed cognitive functioning (26%), weight gain (16%), and mood disorders (16%) were the most common adverse effects. Subsequent analysis showed a better activity for lorlatinib in the presence of K1991E or S1986F mutations, while a limited activity when G2032R mutation occurs. PFROST, a phase II prospective trial evaluating lorlatinib in crizotinib-resistant ROS1-positive NSCLC, confirmed the failure of lorlatinib in the presence of the most common secondary mutations, such as G2032R [66]. Some resistance mechanisms to lorlatinib have been identified, including MET amplification, punctual mutations in the kinase domain, particularly G2032K or L2086F mutations, KRAS mutations, KRAS amplifications, NRAS mutations, and MAP2K1 mutations [16].

## 7. Cabozantinib

Cabozantinib is a small multitarget TKI, binding MET, VEGFR-2, ROS1, RET and AXL [38,43,67]. Preclinical data suggest its capacity to reverse acquired crizotinib-resistance even when mutations such as D2033N or G2032R occur. Its role in TKI (crizotinib, ceritinib, entrectinib)-resistant NSCLC is under investigation. A phase II study evaluating cabozantinib in NSCLC with RET, ROS1, or NTRK fusions or increased AXL and MET activity is ongoing.

## 8. Brigatinib

Brigatinib is an oral, potent, and selective ALK and ROS1 tyrosine kinase inhibitor, approved by the Food and Drug Administration (FDA) and European Medicine Agency (EMA) for patients with ALK positive NSCLC in progression on first-line crizotinib, and more recently for untreated patients. A small trial, conducted on eight patients, has explored brigatinib activity in ROS1-positive NSCLC, reporting an ORR of 37% for the overall population and an ORR of 29% for crizotinib-pretreated patients [68,69]. The lack of activity against the most common ROS1-mutations limited its clinical use.

## 9. Repotrectinib

Repotrectinib, a next-generation TKI, targets ROS-1, ALK, and TRK and passes the blood-brain barrier [70]. It inhibits ROS-1 with >90-fold greater potency than crizotinib and has shown clinical and preclinical activity against the ROS1 G2032 mutation [71,72]. In a phase I/II study, TRIDENT-1, repotrectinib showed encouraging overall and intracranial clinical activity in patients with ROS1-positive NSCLC, with a well-tolerated safety profile. In particular, in TKI-naïve patients, 78.9% (95% CI, 67.6–87.7) of patients showed an objective response, with a 12-month landmark DOR of 86.1%. In patients pretreated with 1 prior ROS1 TKI and no prior chemo, an objective response was observed in 37.5% (95% CI, 24.9–51.5) of patients with a 6-month landmark DOR of 79.5%. Intracranial efficacy was observed in both TKI-naïve and TKI-pretreated patients. Repotrectinib safety is well characterized and manageable, allowing for long-term use. Low-grade dizziness, dysgeusia, constipation, and paresthesia are the most common treatment emergent adverse events (TEAEs) [73].

## 10. Taletrectinib

Taletrectinib is a selective inhibitor of ROS-1 and neurotrophic TRK (NTRK) with activity against crizotinib-resistant ROS-1-rearranged NSCLCs, including those harboring the G2032R, L1951R, S1986F, and L2026M mutations [74]. In contrast, its activity seems to be less effective against the D2033N mutation. An American phase I study, conducted on 46 patients, reported an ORR of 33% for patients with crizotinib-resistant NSCLCs. A Japanese phase I trial enrolled 15 patients and showed an ORR of 58.3% for all patients and 66.7% for untreated patients [75,76].

## 11. Ensartinib

Ensartinib is a TKI with in vitro activity against ALK and ROS-1. A phase II trial showed modest efficacy of this molecule (ORR 27%, 95% CI 13.8–44.1) but an interesting brain disease control [77,78].

## 12. Foretinib

Foretinib is a multitarget inhibitor with activity against both native-ROS1 and mutant -ROS1, but its poor tolerability blocked its use in clinical practice [79].

## 13. Other Treatments

Chemotherapy activity has been thoroughly assessed in patients with ROS1-rearranged NSCLC. Several studies found that pemetrexed-based chemotherapy has a high sensitivity for ROS1-positive NSCLC [32,80,81,82]. In these studies, pemetrexed was studied as monotherapy or in combination with a platinum-based drug (with or without bevacizumab), with objective response rates (ORRs) of 45–60% and median progression-free survival (PFS) durations of 5–23 months. In the first-line setting, median PFS was superior with pemetrexed-based versus non-pemetrexed-based chemotherapy. Furthermore, in patients with ROS1-positive NSCLC, pemetrexed-based treatment is more effective than in individuals with other oncogenic drivers (such as KRAS mutations). In one trial, patients with ROS1 fusion-positive NSCLC had a 58% ORR and a 77.5-month median PFS, compared to 30% and 6 months for patients with EGFR-mutant, EML4-ALK fusion-positive, or ROS1, EGFR, ALK, and KRAS wild-type NSCLCs. [32]. Low levels of thymidylate synthase mRNA in ROS1 fusion-positive cancers (relative to ROS1 wild-type cancers) is probably related to the high responsiveness to pemetrexed. Currently, chemotherapy represents the gold-standard in second-line treatment after progression to crizotinib. Other different approaches are under investigation. In particular, the potential role of immunotherapy is not completely clarified. In fact, in in vitro and in vivo models, ROS1 seems to modulate programmed death protein ligand 1 expression (PD-L1) through MEK-ERK and SHP2 signaling pathways, but on the other hand, in clinical practice, ROS1-positive NSCLC do not express PD-L1 and present a low mutation burden [83]. Multiple studies have characterized the immunophenotype of ROS1 fusion-positive NSCLC. In three different studies, three patients had a PD-L1 tumor proportion score of 0%, four had a score of 1–49%, and three had a score of 50%. Tumor mutational loads are generally low (0–5 mutations per megabase) throughout ROS1-positive tumors, although this information has yet to be determined in a large series of ROS1 fusion-positive tumors. The immunotarget registry included seven patients with ROS1 fusion-positive NSCLCs who were treated with single-agent immune-checkpoint inhibitors (ICIs); five of these patients had progressive disease, one had an objective response, and data for the remaining patient were missing (median PFS was not estimated) [84,85]. In patients with ROS1-rearranged NSCLC, ROS1 TKIs or pemetrexed-based chemotherapy should thus be considered before single-agent ICI. Combination therapy with chemo-immunotherapy or between target therapy with antiangiogenic agents is a charming approach to explore [86]. Clinical trials to evaluate efficacy and safety are needed.

## 14. Conclusions

Almost two decades have passed since the discovery of the first ROS1 fusion in NSCLC. During these years, the natural history of this disease has changed thanks to increasing knowledge. In particular, there have been many advances both from a diagnostic and therapeutic point of view. Regarding the diagnostic phase, the next generation sequencing technique allows to test simultaneously many predictive biomarkers, with high sensitivity and specificity, and in particular, identify several ROS1 fusion partners as well as other oncogenic molecular targets. A wider diffusion of the NGS techniques in clinical practice will probably lead to a greater detection of targetable oncogenic drivers, showing that the real prevalence of ROS1 rearrangements is probably higher than we expect from the literature data. Regarding advances in therapy, at least nine TKIs have been studied, two of which have received regulatory approval in multiple countries; Table 4 summarizes the main trials. Additionally, numerous important resistance mechanisms have been identified. Novel TKIs with a higher selectivity and an improved penetration across the blood-brain barrier are emerging and actually under investigation, in order to overcome resistance mechanisms and obtain better results in terms of survival and quality of life. Notably, the results of trials with next-generation TKIs highlight the possible utility of sequential TKI therapy. A better understanding of ROS1 fusion mechanisms could suggest the use of other combination targeting strategies. In the clinic, further studies are needed to study the activity of chemoimmunotherapy or the association of TKIs with chemotherapy and/or immunotherapy in ROS1-rearranged NSCLC. Therefore, it is very important to offer to the patients with a diagnosis of ROS1-positive NSCLC the possibility to take part in clinical trials.

## Figures and Tables

**Figure 1 ijms-24-11495-f001:**
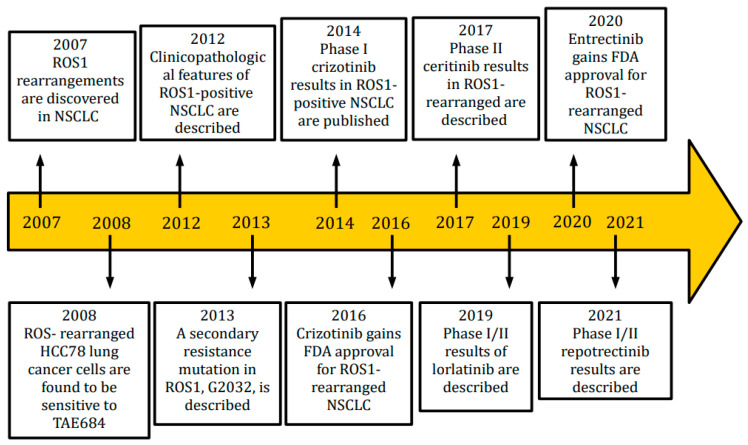
Timeline of main findings in ROS1-positive NSCLC.

**Table 1 ijms-24-11495-t001:** Main prospective clinical trials with crizotinib in ROS1-rearranged NSCLC.

Clinical Trial	Phase	ORR (n)	Median DoR	Median PFS	Median OS
PROFILE 1001 [27,28]	1b	72% (38/53)	24.7 months	19.3 months	51.4 months
EUCROSS [29]	2	70% (21/30)	19 months	15.9 months	20 months
METROS [30]	2	65% (17/26)	21.4 months	22.8 months	-
AcSè [31]	2	69% (25/36)	-	5.5 months	17 months

**Table 2 ijms-24-11495-t002:** Main prospective clinical trials with entrectinib in ROS1-rearranged NSCLC.

Clinical Trial	Phase	ORR	Median TTR	Median TTP	Median PFS	Median OS
STARTRK-1 STARTRK-2 ALKA-372-001 [52]	1/2	67.1%	0.95 months	2.8 months	15.7 months	Not reached
STARTRK-NG [53]	1/2	57.7%	-	-	Not reached	-

**Table 3 ijms-24-11495-t003:** Comparison of major outcomes for crizotinib and entrectinib.

Clinical Trial	Phase	ORR	Median PFS	Median OS
PROFILE 1001 [27,28]	1b	72%	19.3 months	51.4 months
EUCROSS [29]	2	70%	15.9 months	20 months
METROS [30]	2	65%	22.8 months	-
AcSè [31]	2	69%	5.5 months	17 months
STARTRK-1 STARTRK-2 ALKA-372-001 [52]	1/2	67.1%	15.7 months	Not reached
STARTRK-NG [53]	1/2	57.7%	Not reached	-

**Table 4 ijms-24-11495-t004:** Results of main clinical trials in ROS1-rearranged NSCLC.

Clinical Trial	Drug	Phase	ORR	Median PFS	Median OS
PROFILE 1001 [27,28]	Crizotinib	1b	72%	19.3 months	51.4 months
EUCROSS [29]	Crizotinib	2	70%	15.9 months	20 months
METROS [30]	Crizotinib	2	65%	22.8 months	NR
AcSè [31]	Crizotinib	2	69%	5.5 months	17 months
STARTRK-1 STARTRK-2 ALKA-372-001 [52]	Entrectinib	1/2	67.1%	15.7 months	NR
ASCEND-5 [59]	Ceritinib	2	62%	9.3 months (19.3 in crizotinib-naïve pts)	24 months
NCT 01970865 [64,65]	Lorlatinib	1/2	62% in TKI-naïve pts 35% in crizotinib-pretreated pts	21 months in TKI-naïve pts 8.5 months in crizotinib-pretreated pts	-
Dudnik et al. [69]	Brigatinib	1	37% (29% in crizotinib-pretreated pts)	-	-
TRIDENT-1 [73]	Repotrectinib	1/2	78.9% in TKI-naïve pts 37.5% in pretreated pts	-	-
United States and Japan [75,76]	Taletrectinib	1	58.3% (66.7% in TKI-naïve pts)	4 months	-
NCT03608007 [77,78]	Ensartinib	2	27%	-	-

## Data Availability

Not applicable.

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
