# Peer review of "Therapeutical Options in ROS1—Rearranged Advanced Non Small Cell Lung Cancer"

_ijms, 2023, doi:10.3390/ijms241411495_

Round 1

Reviewer 1 Report

The manuscript „Therapeutical Options in ROS1 – rearranged Advanced Non Small Cell Lung Cancer“ by Stanzione and coauthors gives an overview of the therapeutic options for ROS1-positive NSCLC focusing on TKI targeted therapy. The topic is very interesting and, in my opinion, is of great scientific interest. The review is written in a clear way and is nicely organized thus very easy to follow. In addition, the review is detailed, but very focused and I believe will contribute greatly to the scientific field.

However, before accepting the manuscript for publication several minor remarks that concern reference details need to be addressed, such as:

The statement in lines 133-136 lacks the relevant/supporting reference

The authors should also provide the reference that will support the explanation that the transformation of NSCLC to SCLC is one of the mechanisms of resistance to crizotinib (lines 147-151)

At the end of the sentence Drilon et al (line 165) please add reference no. 51.

In paragraph 6. Lorlatinib (lines 239-262), references 64 and 65 are missing in the text.

Reference 32 in the reference list is missing

The references 10 and 91 in the reference list refer to the same article. The reference 91 is redundant/not cited in the text and should be deleted from the reference list.

In the text (line 347) the reference 88 seems to be misplaced

Reviewer 2 Report

Authors submitted manuscript titled “Therapeutical Options in ROS1 – rearranged Advanced Non-Small Cell Lung Cancer” to International Journal of Molecular Sciences. In the review manuscript authors described options how to treat ROS1 positive NSCLC.

Authors in details described current and future options in the treatment of ROS1 positive NSCLC.

However, there are minor suggestions before accepting the manuscript.

1.       Please explain in more details in the introduction section why you stated that squamous cell lung cancers should be tested for ROS1

2.       Please change order of diagnostic tests in the introduction section. IHC, FISH, PCR, NGS is suggested

3.       Explanation of Figure 1 should be in more details in the text

4.       Can you please discuss why in AcSe study is PFS significantly shorter then in other trials with crizotinib

5.       Put reference numbers in Table 1 with the studies

6.       Make table with studies done with entrectinib just like you did it with crizotinib studies

7.       It would be much easier if you could provide a table for comparison of major outcomes for crizotinib and entrectinib are described

8.       Please make a table in the conclusion section with comparison of all mentioned drugs with most important outcomes (ORR, DOR, PFS, OS…..)
